# Fully Parameterized Quantile Function for Distributional Reinforcement Learning

**Derek Yang**[*]
UC San Diego
dyang1206@gmail.com

**Li Zhao**
Microsoft Research
lizo@microsoft.com

**Zichuan Lin**
Tsinghua University
linzc16@mails.tsinghua.edu.cn

**Tao Qin**
Microsoft Research
taoqin@microsoft.com

**Jiang Bian**
Microsoft Research
jiang.bian@microsoft.com

**Tieyan Liu**
Microsoft Research
tyliu@microsoft.com

## Abstract

Distributional Reinforcement Learning (RL) differs from traditional RL in that, rather than the expectation of total returns, it estimates distributions and has achieved state-of-the-art performance on Atari Games. The key challenge in practical distributional RL algorithms lies in how to parameterize estimated distributions so as to better approximate the true continuous distribution. Existing distributional RL algorithms parameterize either the probability side or the return value side of the distribution function, leaving the other side uniformly fixed as in C51, QR-DQN or randomly sampled as in IQN. In this paper, we propose fully parameterized quantile function that parameterizes both the quantile fraction axis (i.e., the x-axis) and the value axis (i.e., y-axis) for distributional RL. Our algorithm contains a fraction proposal network that generates a discrete set of quantile fractions and a quantile value network that gives corresponding quantile values. The two networks are jointly trained to find the best approximation of the true distribution. Experiments on 55 Atari Games show that our algorithm significantly outperforms existing distributional RL algorithms and creates a new record for the Atari Learning Environment for non-distributed agents.

## 1   Introduction

Distributional reinforcement learning [Jaquette et al., 1973, Sobel, 1982, White, 1988, Morimura et al., 2010, Bellemare et al., 2017] differs from value-based reinforcement learning in that, instead of focusing only on the expectation of the return, distributional reinforcement learning also takes the intrinsic randomness of returns within the framework into consideration [Bellemare et al., 2017, Dabney et al., 2018b,a, Rowland et al., 2018]. The randomness comes from both the environment itself and agent's policy. Distributional RL algorithms characterize the total return as random variable and estimate the distribution of such random variable, while traditional Q-learning algorithms estimate only the mean (i.e., traditional value function) of such random variable.

The main challenge of distributional RL algorithm is how to parameterize and approximate the distribution. In Categorical DQN [Bellemare et al., 2017](C51), the possible returns are limited to a discrete set of fixed values, and the probability of each value is learned through interacting with environments. C51 out-performs all previous variants of DQN on a set of 57 Atari 2600 games in the Arcade Learning Environment (ALE) [Bellemare et al., 2013]. Another approach for distributional reinforcement learning is to estimate the quantile values instead. Dabney et al. [2018b] proposed QR-

---

[*]Contributed during internship at Microsoft Research.

DQN to compute the return quantiles on fixed, uniform quantile fractions using quantile regression and minimize the quantile Huber loss [Huber, 1964] between the Bellman updated distribution and current return distribution. Unlike C51, QR-DQN has no restrictions or bound for value and achieves significant improvements over C51. However, both C51 and QR-DQN approximate the distribution function or quantile function on fixed locations, either value or probability. Dabney et al. [2018a] propose learning the quantile values for sampled quantile fractions rather than fixed ones with an implicit quantile value network (IQN) that maps from quantile fractions to quantile values. With sufficient network capacity and infinite number of quantiles, IQN is able to approximate the full quantile function.

However, it is impossible to have infinite quantiles in practice. With limited number of quantile fractions, efficiency and effectiveness of the samples must be reconsidered. The sampling method in IQN mainly helps training the implicit quantile value network rather than approximating the full quantile function, and thus there is no guarantee in that sampled probabilities would provide better quantile function approximation than fixed probabilities.

In this work, we extend the method in Dabney et al. [2018b] and Dabney et al. [2018a] and propose to fully parameterize the quantile function. By fully parameterization, we mean that unlike QR-DQN and IQN where quantile fractions are fixed or sampled and only the corresponding quantile values are parameterized, both quantile fractions and corresponding quantile values in our algorithm are parameterized. In addition to a quantile value network similar to IQN that maps quantile fractions to corresponding quantile values, we propose a fraction proposal network that generates quantile fractions for each state-action pair. The fraction proposal network is trained so that as the true distribution is approximated, the 1-Wasserstein distance between the approximated distribution and the true distribution is minimized. Given the proposed fractions generated by the fraction proposal network, we can learn the quantile value network by quantile regression. With self-adjusting fractions, we can approximate the true distribution better than with fixed or sampled fractions.

We begin with related works and backgrounds of distributional RL in Section 2. We describe our algorithm in Section 3 and provide experiment results of our algorithm on the ALE environment [Bellemare et al., 2013] in Section 4. At last, we discuss the future extension of our work, and conclude our work in Section 5.

## 2  Background and Related Work

We consider the standard reinforcement learning setting where agent-environment interactions are modeled as a Markov Decision Process $(\mathcal{X}, \mathcal{A}, R, P, \gamma)$ [Puterman, 1994], where $\mathcal{X}$ and $\mathcal{A}$ denote state space and action space, $P$ denotes the transition probability given state and action, $R$ denotes state and action dependent reward function and $\gamma \in (0, 1)$ denotes the reward discount factor.

For a policy $\pi$, define the discounted return sum a random variable by $Z^\pi(x, a) = \sum_{t=0}^\infty \gamma^t R(x_t, a_t)$, where $x_0 = x$, $a_0 = a$, $x_t \sim P(\cdot|x_{t-1}, a_{t-1})$ and $a_t \sim \pi(\cdot|x_t)$. The objective in reinforcement learning can be summarized as finding the optimal $\pi^*$ that maximizes the expectation of $Z^\pi$, the action-value function $Q^\pi(x, a) = \mathbb{E}[Z^\pi(x, a)]$. The most common approach is to find the unique fixed point of the Bellman optimality operator $\mathcal{T}$ [Bellman, 1957]:

$$Q^*(x, a) = \mathcal{T}Q^*(x, a) := \mathbb{E}[R(x, a)] + \gamma \mathbb{E}_P \max_{a'} Q^*(x', a').$$

To update $Q$, which is approximated by a neural network in most deep reinforcement learning studies, $Q$-learning [Watkins, 1989] iteratively trains the network by minimizing the squared temporal difference (TD) error defined by

$$\delta_t^2 = \left[ r_t + \gamma \max_{a' \in \mathcal{A}} Q(x_{t+1}, a') - Q(x_t, a_t) \right]^2$$

along the trajectory observed while the agent interacts with the environment following $\epsilon$-greedy policy. DQN [Mnih et al., 2015] uses a convolutional neural network to represent $Q$ and achieves human-level play on the Atari-57 benchmark.

## 2.1 Distributional RL

Instead of a scalar $Q^\pi(x, a)$, distributional RL looks into the intrinsic randomness of $Z^\pi$ by studying its distribution. The distributional Bellman operator for policy evaluation is

$$Z^\pi(x, a) \stackrel{D}{=} R(x, a) + \gamma Z^\pi(X', A'),$$

where $X' \sim P(\cdot|x, a)$ and $A' \sim \pi(\cdot|X')$, $A \stackrel{D}{=} B$ denotes that random variable $A$ and $B$ follow the same distribution.

Both theory and algorithms have been established for distributional RL. In theory, the distributional Bellman operator for policy evaluation is proved to be a contraction in the $p$-Wasserstein distance [Bellemare et al., 2017]. Bellemare et al. [2017] shows that C51 outperforms value-based RL, in addition Hessel et al. [2018] combined C51 with enhancements such as prioritized experience replay [Schaul et al., 2016], n-step updates [Sutton, 1988], and the dueling architecture [Wang et al., 2016], leading to the Rainbow agent, current state-of-the-art in Atari-57 for non-distributed agents, while the distributed algorithm proposed by Kapturowski et al. [2018] achieves state-of-the-art performance for all agents. From an algorithmic perspective, it is impossible to represent the full space of probability distributions with a finite collection of parameters. Therefore the parameterization of quantile functions is usually the most crucial part in a general distributional RL algorithm. In C51, the true distribution is projected to a categorical distribution [Bellemare et al., 2017] with fixed values for parameterization. QR-DQN fixes probabilities instead of values, and parameterizes the quantile values [Dabney et al., 2018a] while IQN randomly samples the probabilities [Dabney et al., 2018a]. We will introduce QR-DQN and IQN in Section 2.2, and extend from their work to ours.

## 2.2 Quantile Regression for Distributional RL

In contrast to C51 which estimates probabilities for $N$ fixed locations in return, QR-DQN [Dabney et al., 2018b] estimates the respected quantile values for $N$ fixed, uniform probabilities. In QR-DQN, the distribution of the random return is approximated by a uniform mixture of $N$ Diracs,

$$Z_\theta(x, a) := \frac{1}{N} \sum_{i=1}^{N} \delta_{\theta_i(x,a)},$$

with each $\theta_i$ assigned a quantile value trained with quantile regression.

Based on QR-DQN, Dabney et al. [2018a] propose using probabilities sampled from a base distribution, e.g. $\tau \in U([0, 1])$, rather than fixed probabilities. They further learn the quantile function that maps from embeddings of sampled probabilities to the corresponding quantiles, called implicit quantile value network (IQN). At the time of this writing, IQN achieves the state-or-the-art performance on Atari-57 benchmark, human-normalized mean and median of all agents that does not combine distributed RL, prioritized replay [Schaul et al., 2016] and $n$-step update.

Dabney et al. [2018a] claimed that with enough network capacity, IQN is able to approximate to the full quantile function with infinite number of quantile fractions. However, in practice one needs to use a finite number of quantile fractions to estimate action values for decision making, e.g. 32 randomly sampled quantile fractions as in Dabney et al. [2018a]. With limited fractions, a natural question arises that, how to best utilize those fractions to find the closest approximation of the true distribution?

## 3 Our Algorithm

We propose Fully parameterized Quantile Function (FQF) for Distributional RL. Our algorithm consists of two networks, the fraction proposal network that generates a set of quantile fractions for each state-action pair, and the quantile value network that maps probabilities to quantile values. We first describe the fully parameterized quantile function in Section 3.1, with variables on both probability axis and value axis. Then, we show how to train the fraction proposal network in Section 3.2, and how to train the quantile value network with quantile regression in Section 3.3. Finally, we present our algorithm and describe the implementation details in Section 3.4.

## 3.1 Fully Parameterized Quantile Function

In FQF, we estimate $N$ adjustable quantile values for $N$ adjustable quantile fractions to approximate the quantile function. The distribution of the return is approximated by a weighted mixture of $N$ Diracs given by

$$Z_{\theta,\tau}(x,a) := \sum_{i=0}^{N-1} (\tau_{i+1} - \tau_i)\delta_{\theta_i(x,a)}, \tag{1}$$

where $\delta_z$ denotes a Dirac at $z \in \mathbb{R}$, $\tau_1, ... \tau_{N-1}$ represent the N-1 adjustable fractions satisfying $\tau_{i-1} < \tau_i$, with $\tau_0 = 0$ and $\tau_N = 1$ to simplify notation. Denote quantile function [Müller, 1997] $F_Z^{-1}$ the inverse function of cumulative distribution function $F_Z(z) = Pr(Z < z)$. By definition we have

$$F_Z^{-1}(p) := \inf \{z \in \mathbb{R} : p \le F_Z(z)\}$$

where $p$ is what we refer to as quantile fraction.

Based on the distribution in Eq.(1), denote $\Pi^{\theta,\tau}$ the projection operator that projects quantile function onto a staircase function supported by $\theta$ and $\tau$, the projected quantile function is given by

$$F_Z^{-1,\theta,\tau}(\omega) = \Pi^{\theta,\tau} F_Z^{-1}(\omega) = \theta_0 + \sum_{i=0}^{N-1}(\theta_{i+1} - \theta_i)H_{\tau_{i+1}}(\omega),$$

where $H$ is the Heaviside step function and $H_\tau(\omega)$ is the short for $H(\omega - \tau)$. Figure 1 gives an example of such projection. For each state-action pair $(x,a)$, we first generate the set of fractions $\tau$ using the fraction proposal network, and then obtain the quantiles values $\theta$ corresponding to $\tau$ using the quantile value network.

To measure the distortion between approximated quantile function and the true quantile function, we use the 1-Wasserstein metric given by

$$W_1(Z,\theta,\tau) = \sum_{i=0}^{N-1} \int_{\tau_i}^{\tau_{i+1}} \left| F_Z^{-1}(\omega) - \theta_i \right| d\omega. \tag{2}$$

Unlike KL divergence used in C51 which considers only the probabilities of the outcomes, the $p$-Wasseretein metric takes both the probability and the distance between outcomes into consideration. Figure 1 illustrates the concept of how different approximations could affect $W_1$ error, and shows an example of $\Pi_{W_1}$. However, note that in practice Eq.(2) can not be obtained without bias.

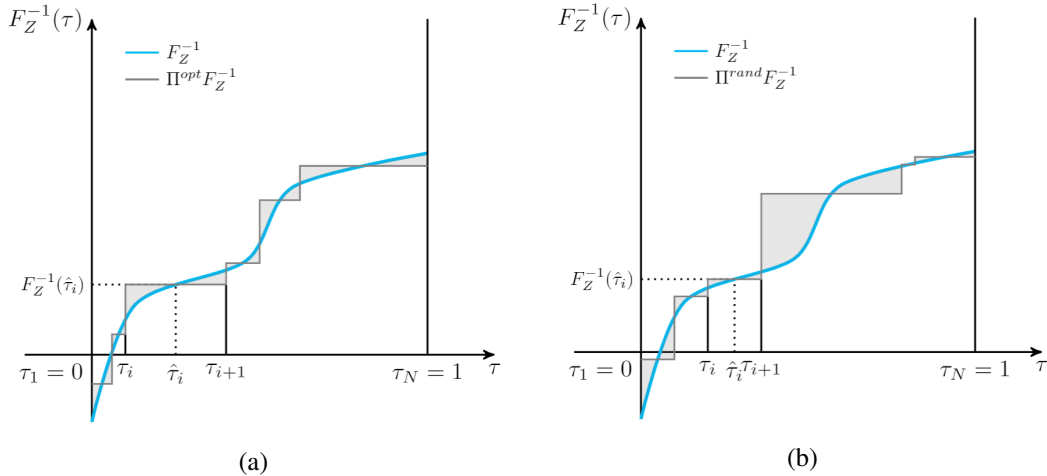

(a)                                                                                        (b)

Figure 1: Two approximations of the same quantile function using different set of $\tau$ with $N = 6$, the area of the shaded region is equal to the 1-Wasserstein error. (a) Finely-adjusted $\tau$ with minimized $W_1$ error. (b) Randomly chosen $\tau$ with larger $W_1$ error.

## 3.2 Training fraction proposal Network

To achieve minimal 1-Wasserstein error, we start from fixing $\tau$ and finding the optimal corresponding quantile values $\theta$. In QR-DQN, Dabney et al. [2018a] gives an explicit form of $\theta$ to achieve the goal. We extend it to our setting:

**Lemma 1.** *[Dabney et al., 2018a] For any $\tau_1, ... \tau_{N-1} \in [0, 1]$ satisfying $\tau_{i-1} < \tau_i$ for $i$, with $\tau_1 = 0$ and $\tau_N = 1$, and cumulative distribution function $F$ with inverse $F^{-1}$, the set of $\theta$ minimizing Eq.(2) is given by*

$$\theta_i = F_Z^{-1}(\frac{\tau_i + \tau_{i+1}}{2}) \tag{3}$$

We can now substitute $\theta_i$ in Eq.(2) with equation Eq.(3) and find the optimal condition for $\tau$ to minimize $W_1(Z, \tau)$. For simplicity, we denote $\hat{\tau}_i = \frac{\tau_i + \tau_{i+1}}{2}$.

**Proposition 1.** *For any continuous quantile function $F_Z^{-1}$ that is non-decreasing, define the 1-Wasserstein loss of $F_Z^{-1}$ and $F_Z^{-1,\tau}$ by*

$$W_1(Z, \tau) = \sum_{i=0}^{N-1} \int_{\tau_i}^{\tau_{i+1}} \left| F_Z^{-1}(\omega) - F_Z^{-1}(\hat{\tau}_i) \right| d\omega. \tag{4}$$

$\frac{\partial W_1}{\partial \tau_i}$ *is given by*

$$\frac{\partial W_1}{\partial \tau_i} = 2F_Z^{-1}(\tau_i) - F_Z^{-1}(\hat{\tau}_i) - F_Z^{-1}(\hat{\tau}_{i-1}), \tag{5}$$

$\forall i \in (0, N)$.

*Further more, $\forall \tau_{i-1}, \tau_{i+1} \in [0, 1], \tau_{i-1} < \tau_{i+1}, \exists \tau_i \in (\tau_{i-1}, \tau_{i+1})$ s.t. $\frac{\partial W_1}{\partial \tau_i} = 0$.*

Proof of proposition 1 is given in the appendix. While computing $W_1$ without bias is usually impractical, equation 5 provides us with a way to minimize $W_1$ without computing it. Let $w_1$ be the parameters of the fraction proposal network $P$, for an arbitrary quantile function $F_Z^{-1}$, we can minimize $W_1$ by iteratively applying gradients descent to $w_1$ according to Eq.(5) and convergence is guaranteed. As the true quantile function $F_Z^{-1}$ is unknown to us in practice, we use the quantile value network $F_{Z,w_2}^{-1}$ with parameters $w_2$ for current state and action as true quantile function.

The expected return, also known as action-value based on FQF is then given by

$$Q(x, a) = \sum_{i=0}^{N-1} (\tau_{i+1} - \tau_i) F_{Z,w_2}^{-1}(\hat{\tau}_i),$$

where $\tau_0 = 0$ and $\tau_N = 1$.

## 3.3 Training quantile value network

With the properly chosen probabilities, we combine quantile regression and distributional Bellman update on the optimized probabilities to train the quantile function. Consider $Z$ a random variable denoting the action-value at $(x_t, a_t)$ and $Z'$ the action-value random variable at $(x_{t+1}, a_{t+1})$, the weighted temporal difference (TD) error for two probabilities $\hat{\tau}_i$ and $\hat{\tau}_j$ is defined by

$$\delta_{ij}^t = r_t + \gamma F_{Z',w_1}^{-1}(\hat{\tau}_i) - F_{Z,w_1}^{-1}(\hat{\tau}_j) \tag{6}$$

Quantile regression is used in QR-DQN and IQN to stochastically adjust the quantile estimates so as to minimize the Wasserstein distance to a target distribution. We follow QR-DQN and IQN where quantile value networks are trained by minimizing the Huber quantile regression loss [Huber, 1964], with threshold $\kappa$,

$$\rho_\tau^\kappa(\delta_{ij}) = |\tau - \mathbb{I}\{\delta_{ij} < 0\}| \frac{\mathcal{L}_\kappa(\delta_{ij})}{\kappa}, \text{ with}$$

$$\mathcal{L}_\kappa(\delta_{ij}) = \begin{cases} \frac{1}{2}\delta_{ij}^2, & \text{if } |\delta_{ij}| \leq \kappa \\ \kappa\left(|\delta_{ij}| - \frac{1}{2}\kappa\right), & \text{otherwise} \end{cases}$$

The loss of the quantile value network is then given by

$$\mathcal{L}(x_t, a_t, r_t, x_{t+1}) = \frac{1}{N} \sum_{i=0}^{N-1} \sum_{j=0}^{N-1} \rho_{\hat{\tau}_j}^{\kappa}(\delta_{ij}^t) \tag{7}$$

Note that $F_Z^{-1}$ and its Bellman target share the same proposed quantile fractions $\hat{\tau}$ to reduce computation.

We perform joint gradient update for $w_1$ and $w_2$, as illustrated in Algorithm 1.

---

**Algorithm 1:** FQF update

---

**Parameter :** $N, \kappa$
**Input:** $x, a, r, x', \gamma \in [0, 1)$
// Compute proposed fractions for $x, a$
$\tau \leftarrow P_{w_1}(x)$;
// Compute proposed fractions for $x', a'$
**for** $a' \in \mathcal{A}$ **do**
   |   $\tau' \leftarrow P_{w_1}(x')$;
**end**
// Compute greedy action
$Q(s', a') \leftarrow \sum_{i=0}^{N-1} (\tau'_{i+1} - \tau'_i) F_{Z', w_2}^{-1}(\hat{\tau}_i)_a$;
$a^* \leftarrow \underset{a'}{\operatorname{argmax}} Q(s', a')$;
// Compute $L$
**for** $0 \le i \le N - 1$ **do**
   |   **for** $0 \le j \le N - 1$ **do**
   |    |   $\delta_{ij} \leftarrow r + \gamma F_{Z', w_2}^{-1}(\hat{\tau}_i) - F_{Z, w_2}^{-1}(\hat{\tau}_j)$
   |   **end**
**end**
$\mathcal{L} = \frac{1}{N} \sum_{i=0}^{N-1} \sum_{j=0}^{N-1} \rho_{\hat{\tau}_j}^{\kappa}(\delta_{ij})$;
// Compute $\frac{\partial W_1}{\partial \tau_i}$ for $i \in [1, N-1]$
$\frac{\partial W_1}{\partial \tau_i} = 2 F_{Z, w_2}^{-1}(\tau_i) - F_{Z, w_2}^{-1}(\hat{\tau}_i) - F_{Z, w_2}^{-1}(\hat{\tau}_{i-1})$;
Update $w_1$ with $\frac{\partial W_1}{\partial \tau_i}$; Update $w_2$ with $\nabla \mathcal{L}$;
**Output:** $Q$

---

### 3.4 Implementation Details

Our fraction proposal network is represented by one fully-connected MLP layer. It takes the state embedding of original IQN as input and generates fraction proposal. Recall that in Proposition 1, we require $\tau_{i-1} < \tau_i$ and $\tau_0 = 0, \tau_N = 1$. While it is feasible to have $\tau_0 = 0, \tau_N = 1$ fixed and sort the output of $\tau_{w_1}$, the sort operation would make the network hard to train. A more reasonable and practical way would be to let the neural network automatically have the output sorted using cumulated softmax. Let $q \in \mathbb{R}^N$ denote the output of a softmax layer, we have $q_i \in (0, 1), i \in [0, N-1]$ and $\sum_{i=0}^{N-1} q_i = 1$. Let $\tau_i = \sum_{j=0}^{i-1} q_j, i \in [0, N]$, then straightforwardly we have $\tau_i < \tau_j$ for $\forall i < j$ and $\tau_0 = 0, \tau_N = 1$ in our fraction proposal network. Note that as $W_1$ is not computed, we can't directly perform gradient descent for the fraction proposal network. Instead, we use the `grad_ys` argument in the tensorflow operator `tf.gradients` to assign $\frac{\partial W_1}{\partial \tau_i}$ to the optimizer. In addition, one can use entropy of $q$ as a regularization term $H(q) = -\sum_{i=0}^{N-1} q_i \log q_i$ to prevent the distribution from degenerating into a deterministic one.

We borrow the idea of implicit representations from IQN to our quantile value network. To be specific, we compute the embedding of $\tau$, denoted by $\phi(\tau)$, with

$$\phi_j(\tau) := \text{ReLU}\left(\sum_{i=0}^{n-1} \cos(i\pi\tau) w_{ij} + b_j\right),$$

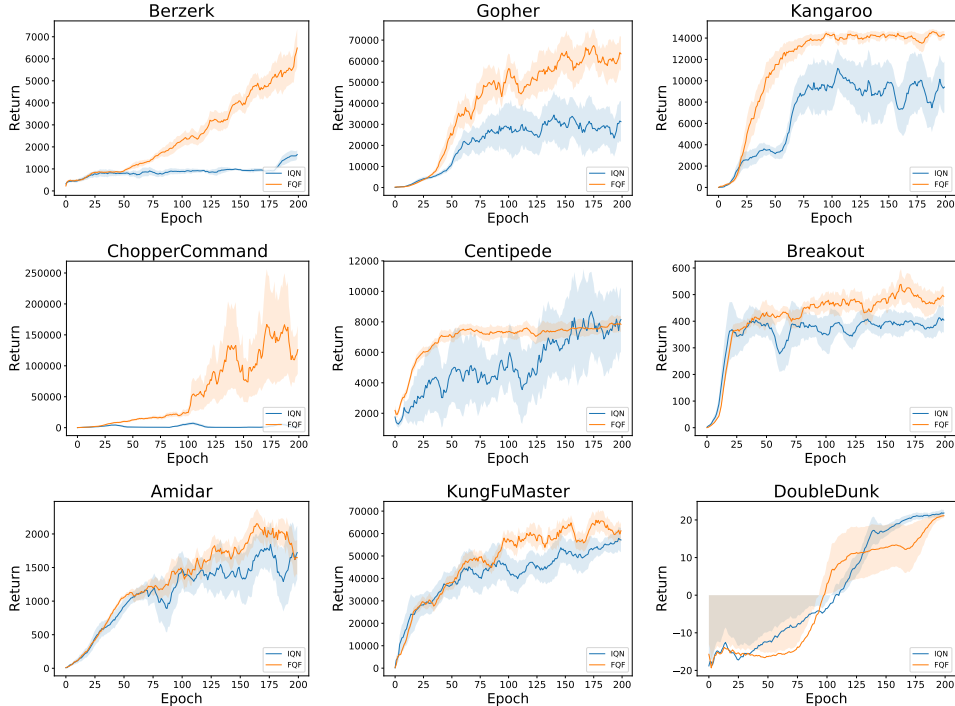

Figure 2: Performance comparison with IQN. Each training curve is averaged by 3 seeds. The training curves are smoothed with a moving average of 10 to improve readability.

where $w_{ij}$ and $b_j$ are network parameters. We then compute the element-wise (Hadamard) product of state feature $\psi(x)$ and embedding $\phi(\tau)$. Let $\odot$ denote element-wise product, the quantile values are given by $F_Z^{-1}(\tau) \approx F_{Z,w_2}^{-1}(\psi(x) \odot \phi(\tau))$.

In IQN, after the set of $\tau$ is sampled from a uniform distribution, instead of using differences between $\tau$ as probabilities of the quantiles, the mean of the quantile values is used to compute action-value $Q$. While in expectation, $Q = \sum_{i=0}^{N-1}(\tau_{i+1} - \tau_i)F_Z^{-1}(\frac{\tau_i+\tau_{i+1}}{2})$ with $\tau_0 = 0, \tau_N = 1$ and $Q = \frac{1}{N}\sum_{i=1}^{N} F_Z^{-1}(\tau_i)$ are equal, we use the former one to consist with our projection operation.

## 4 Experiments

We test our algorithm on the Atari games from Arcade Learning Environment (ALE) Bellemare et al. [2013]. We select the most relative algorithm to ours, IQN [Dabney et al., 2018a], as baseline, and compare FQF with QR-DQN [Dabney et al., 2018b], C51 [Bellemare et al., 2017], prioritized experience replay [Schaul et al., 2016] and Rainbow [Hessel et al., 2018], the current state-of-art that combines the advantages of several RL algorithms including distributional RL. The baseline algorithm is implemented by Castro et al. [2018] in the Dopamine framework, with slightly lower performance than reported in IQN. We implement FQF based on the Dopamine framework. Unfortunately, we fail to test our algorithm on *Surround* and *Defender* as *Surround* is not supported by the Dopamine framework and scores of *Defender* is unreliable in Dopamine. Following the common practice [Van Hasselt et al., 2016], we use the 30-noop evaluation settings to align with previous works. Results of FQF and IQN using sticky action for evaluation proposed by Machado et al. [2018] are also provided in the appendix. In all, the algorithms are tested on 55 Atari games.

Our hyper-parameter setting is aligned with IQN for fair comparison. The number of $\tau$ for FQF is 32. The weights of the fraction proposal network are initialized so that initial probabilities are uniform as in QR-DQN, also the learning rates are relatively small compared with the quantile value network to keep the probabilities relatively stable while training. We run all agents with 200 million frames. At the training stage, we use $\epsilon$-greedy with $\epsilon = 0.01$. For each evaluation stage, we test the agent for

0.125 million frames with $\epsilon = 0.001$. For each algorithm we run 3 random seeds. All experiments are performed on NVIDIA Tesla V100 16GB graphics cards.

|         | Mean   | Median | >Human | >DQN |
|---------|--------|--------|--------|------|
| DQN     | 221%   | 79%    | 24     | 0    |
| PRIOR.  | 580%   | 124%   | 39     | 48   |
| C51     | 701%   | 178%   | 40     | 50   |
| RAINBOW | 1213%  | 227%   | 42     | 52   |
| QR-DQN  | 902%   | 193%   | 41     | 54   |
| IQN     | 1112%  | 218%   | 39     | 54   |
| FQF     | **1426**% | **272**% | **44** | **54** |

Table 1: Mean and median scores across 55 Atari 2600 games, measured as percentages of human baseline. Scores are averages over 3 seeds.

Table 1 compares the mean and median human normalized scores across 55 Atari games with up to 30 random no-op starts, and the full score table is provided in the Appendix. It shows that FQF outperforms all existing distributional RL algorithms, including Rainbow [Hessel et al., 2018] that combines C51 with prioritized replay, and n-step updates. We also set a new record on the number of games where non-distributed RL agent performs better than human.

Figure 2 shows the training curves of several Atari games. Even on games where FQF and IQN have similar performance such as *Centipede*, FQF is generally much faster thanks to self-adjusting fractions.

However, one side effect of the full parameterization in FQF is that the training speed is decreased. With same settings, FQF is roughly 20% slower than IQN due to the additional fraction proposal network. As the number of $\tau$ increases, FQF slows down significantly while IQN's training speed is not sensitive to the number of $\tau$ samples.

## 5   Discussion and Conclusions

Based on previous works of distributional RL, we propose a more general complete approximation of the return distribution. Compared with previous distributional RL algorithms, FQF focuses not only on learning the target, e.g. probabilities for C51, quantile values for QR-DQN and IQN, but also which target to learn, i.e quantile fraction. This allows FQF to learn a better approximation of the true distribution under restrictions of network capacity. Experiment result shows that FQF does achieve significant improvement.

There are some open questions we are yet unable to address in this paper. We will have some discussions here. First, does the 1-Wasserstein error converge to its minimal value when the quantile function is not fixed? We cannot guarantee convergence of the fraction proposal network in deep neural networks where we involve quantile regression and Bellman update. Second, though we empirically believe so, does the contraction mapping result for fixed probabilities given by Dabney et al. [2018b] also apply on self-adjusting probabilities? Third, while FQF does provide potentially better distribution approximation with same amount of fractions, how will a better approximated distribution affect agent's policy and how will it affect the training process? More generally, how important is quantile fraction selection during training?

As for future work, we believe that studying the trained quantile fractions will provide intriguing results. Such as how sensitive are the quantile fractions to state and action, and that how the quantile fractions will evolve in a single run. Also, the combination of distributional RL and DDPG in D4PG [Barth-Maron et al., 2018] showed that distributional RL can also be extended to continuous control settings. Extending our algorithm to continuous settings is another interesting topic. Furthermore, in our algorithm we adopted the concept of selecting the best target to learn. Can this intuition be applied to areas other than RL?

Finally, we also noticed that most of the games we fail to reach human-level performance involves complex rules that requires exploration based policies, such as *Montezuma Revenge* and *Venture*. Integrating distributional RL will be another potential direction as in [Tang and Agrawal, 2018]. In

general, we believe that our algorithm can be viewed as a natural extension of existing distributional RL algorithms, and that distributional RL may integrate greatly with other algorithms to reach higher performance.

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
