[Supplementary Material]

# Appendix

## Proof for proposition 1

**Proposition 1.** *For any continuous quantile function $F_Z^{-1}$ that is non-decreasing, define the 1-Wasserstein loss of $F_Z^{-1}$ and $F_Z^{-1,\tau}$ by*

$$W_1(Z,\tau) = \sum_{i=0}^{N-1} \int_{\tau_i}^{\tau_{i+1}} \left| F_Z^{-1}(\omega) - F_Z^{-1}(\hat{\tau}_i) \right| d\omega. \tag{4}$$

*$\frac{\partial W_1}{\partial \tau_i}$ is given by*

$$\frac{\partial W_1}{\partial \tau_i} = 2F_Z^{-1}(\tau_i) - F_Z^{-1}(\hat{\tau}_i) - F_Z^{-1}(\hat{\tau}_{i-1}), \tag{5}$$

*$\forall i \in (0, N)$.*

*Further more, $\forall \tau_{i-1}, \ \tau_{i+1} \in [0,1], \ \tau_{i-1} < \tau_{i+1}, \ \exists \tau_i \in (\tau_{i-1}, \tau_{i+1}) \ s.t. \ \frac{\partial W_1}{\partial \tau_i} = 0$.*

*Proof.* Note that $F_Z^{-1}$ is non-decreasing. We have

$$
\begin{aligned}
\frac{\partial W_1}{\partial \tau_i} =& \frac{\partial}{\partial \tau_i} \left( \int_{\tau_{i-1}}^{\tau_i} \left| F_Z^{-1}(\omega) - F_Z^{-1}(\hat{\tau}_{i-1}) \right| d\omega + \int_{\tau_i}^{\tau_{i+1}} \left| F_Z^{-1}(\omega) - F_Z^{-1}(\hat{\tau}_i) \right| d\omega \right) \\
=& \frac{\partial}{\partial \tau_i} \left( \int_{\tau_{i-1}}^{\hat{\tau}_{i-1}} F_Z^{-1}(\hat{\tau}_{i-1}) - F_Z^{-1}(\omega) d\omega + \int_{\hat{\tau}_{i-1}}^{\tau_i} F_Z^{-1}(\omega) - F_Z^{-1}(\hat{\tau}_{i-1}) d\omega + \right. \\
& \left. \int_{\tau_i}^{\tau_{i+1}} \left| F_Z^{-1}(\omega) - F_Z^{-1}(\hat{\tau}_i) \right| d\omega \right) \\
=& \frac{\tau_i - \tau_{i-1}}{4} \frac{\partial}{\partial \tau_i} F_Z^{-1}(\hat{\tau}_{i-1}) + F_Z^{-1}(\tau_i) - F_Z^{-1}(\hat{\tau}_{i-1}) - \frac{\tau_i - \tau_{i-1}}{4} \frac{\partial}{\partial \tau_i} F_Z^{-1}(\hat{\tau}_{i-1}) + \\
& \frac{\partial}{\partial \tau_i} \left( \int_{\tau_i}^{\tau_{i+1}} \left| F_Z^{-1}(\omega) - F_Z^{-1}(\hat{\tau}_i) \right| d\omega \right) \\
=& F_Z^{-1}(\tau_i) - F_Z^{-1}(\hat{\tau}_{i-1}) + \frac{\partial}{\partial \tau_i} \left( \int_{\tau_i}^{\tau_{i+1}} \left| F_Z^{-1}(\omega) - F_Z^{-1}(\hat{\tau}_i) \right| d\omega \right) \\
=& F_Z^{-1}(\tau_i) - F_Z^{-1}(\hat{\tau}_{i-1}) + F_Z^{-1}(\tau_i) - F_Z^{-1}(\hat{\tau}_i) \\
=& 2F_Z^{-1}(\tau_i) - F_Z^{-1}(\hat{\tau}_{i-1}) - F_Z^{-1}(\hat{\tau}_i)
\end{aligned}
$$

As $F_Z^{-1}$ is non-decreasing we have $\frac{\partial W_1}{\partial \tau_i}\big|_{\tau_i=\tau_{i-1}} \leq 0$ and $\frac{\partial W_1}{\partial \tau_i}\big|_{\tau_i=\tau_{i+1}} \geq 0$. Recall that $F_Z^{-1}$ is continuous, so $\exists \tau_i \in (\tau_{i-1}, \tau_{i+1}) \ s.t. \ \frac{\partial W_1}{\partial \tau_i} = 0$. $\square$

**Hyper-parameter sheet**

| Hyper-parameter | IQN | FQF |
|---|---|---|
| Learning rate | 0.00005 | 0.00005 |
| Optimizer | Adam | Adam |
| Batch size | 32 | 32 |
| Discount factor | 0.99 | 0.99 |
| Fraction proposal network learning rate | None | 2.5e-9 |
| Fraction proposal network optimizer | None | RMSProp |

Table 2: hyper-parameter list

We sweep the learning rate of fraction proposal network among (0, 2.5e-5) and finally fix this learning rate as 2.5e-9. For the training of fraction proposal network, we use RMSProp optimizer. Note that though the fraction proposal network takes the state embedding of original IQN as input, we only apply gradient to our new introduced parameter and do not back-propagate the gradient to the convolution layers.

**Approximation demonstration**

To demonstrate how FQF provides a better quantile function approximation, figure 3 provides plots of a toy case with different distributional RL algorithm's approximation of a known quantile function, from which we can see how quantile fraction selection affects distribution approximation.

(a)                                    (b)

Figure 3: Demonstration of quantile function approximation on a toy case. $W_1$ denotes 1-Wasserstein distance between the approximated function and the one obtained through MC method.

**Varying number of quantile fractions**

Table 3 gives mean scores of FQF and IQN over 6 Atari games, using different number of quantile fractions, i.e. $N$. For IQN, the selection of $N'$ is based on the highest score of each column given in *Figure 2* of [Dabney et al., 2018a].

|  | N=8 | N=32 | N=64 |
|---|---|---|---|
| IQN | 60.2 | 91.5 | 64.4 |
| FQF | 83.2 | 124.6 | 69.5 |

Table 3: Mean scores across 6 Atari 2600 games, measured as percentages of human baseline. Scores are averages over 3 seeds.

Intuitively, the advantage of trained quantile fractions compared to random ones will be more observable at smaller $N$. At larger $N$ when both trained quantile fractions and random ones are densely distributed over $[0, 1]$, the differences between FQF and IQN becomes negligible. However from table 3 we see that even at large $N$, FQF performs slightly better than IQN.

**Visualizing proposed quantile fraction**

In figure 4, we select a half-trained *Kungfu Master* agent with $N = 8$ to provide a case study of FQF. The reason why we choose a half-trained agent instead of a fully-trained agent is so that the distribution of $Q$ is not a deterministic one. Note that theoretically the quantile function should be non-decreasing, however from the example we can see that the learned quantile function might not always follow this property, and this phenomenon further motivates a quite interesting future work that leverages the non-decreasing property as prior knowledge for quantile function learning. The figure shows how the interval between proposed quantile fractions (i.e., the output of the softmax layer that sums to 1. See Section 3.4 for details) vary during a single run.

Figure 4: Interval between adjacent proposed quantile fractions for states at each time step in a single run. Different colors refer to different adjacent fractions' intervals, e.g. green curve refers to $\tau_2 - \tau_1$.

Whenever there appears an enemy behind the character, we see a spike in the fraction interval, indicating that proposed fraction is very different from that of following states without enemies. This suggests that the fraction proposal network is indeed state dependent and is able to provide different quantile fractions accordingly.

**ALE Scores**

| GAMES | RANDOM | HUMAN | DQN | PRIOR.DUEL. | QR-DQN | IQN | FQF |
|---|---|---|---|---|---|---|---|
| Alien | 227.8 | 7127.7 | 1620.0 | 3941.0 | 4871.0 | 7022.0 | **16754.6** |
| Amidar | 5.8 | 1719.5 | 978.0 | 2296.8 | 1641.0 | 2946.0 | **3165.3** |
| Assault | 222.4 | 742.0 | 4280.4 | 11477.0 | 22012.0 | **29091.0** | 23020.1 |
| Asterix | 210.0 | 8503.3 | 4359.0 | 375080.0 | 261025.0 | 342016.0 | **578388.5** |
| Asteroids | 719.1 | 47388.7 | 1364.5 | 1192.7 | 4226.0 | 2898.0 | **4553.0** |
| Atlantis | 12850.0 | 29028.1 | 279987.0 | 395762.0 | 971850.0 | **978200.0** | 957920.0 |
| BankHeist | 14.2 | 753.1 | 455.0 | **1503.1** | 1249.0 | 1416.0 | 1259.1 |
| BattleZone | 2360.0 | 37187.5 | 29900.0 | 35520.0 | 39268.0 | 42244.0 | **87928.6** |
| BeamRider | 363.9 | 16926.5 | 8627.5 | 30276.5 | 34821.0 | **42776.0** | 37106.6 |
| Berzerk | 123.7 | 2630.4 | 585.6 | 3409.0 | 3117.0 | 1053.0 | **12422.2** |
| Bowling | 23.1 | 160.7 | 50.4 | 46.7 | 77.2 | 86.5 | **102.3** |
| Boxing | 0.1 | 12.1 | 88.0 | 98.9 | **99.9** | 99.8 | 98.0 |
| Breakout | 1.7 | 30.5 | 385.5 | 366.0 | 742.0 | 734.0 | **854.2** |
| Centipede | 2090.9 | 12017.0 | 4657.7 | 7687.5 | **12447.0** | 11561.0 | 11526.0 |
| ChopperCommand | 811.0 | 7387.8 | 6126.0 | 13185.0 | 14667.0 | 16836.0 | **876460.0** |
| CrazyClimber | 10780.5 | 35829.4 | 110763.0 | 162224.0 | 161196.0 | 179082.0 | **223470.6** |
| DemonAttack | 152.1 | 1971.0 | 12149.4 | 72878.6 | 121551.0 | 128580.0 | **131697.0** |
| DoubleDunk | -18.6 | -16.4 | -6.6 | -12.5 | 21.9 | 5.6 | **22.9** |
| Enduro | 0.0 | 860.5 | 729.0 | 2306.4 | 2355.0 | 2359.0 | **2370.8** |
| FishingDerby | -91.7 | -38.7 | -4.9 | 41.3 | 39.0 | 33.8 | **52.7** |
| Freeway | 0.0 | 29.6 | 30.8 | 33.0 | **34.0** | **34.0** | 33.7 |
| Frostbite | 65.2 | 4334.7 | 797.4 | 7413.0 | 4384.0 | 4324.0 | **16472.9** |
| Gopher | 257.6 | 2412.5 | 8777.4 | 104368.2 | 113585.0 | 118365.0 | **121144.0** |
| Gravitar | 173.0 | 3351.4 | 473.0 | 238.0 | 995.0 | 911.0 | **1406.0** |
| Hero | 1027.0 | 30826.4 | 20437.8 | 21036.5 | 21395.0 | 28386.0 | **30926.2** |
| IceHockey | -11.2 | 0.9 | -1.9 | -0.4 | -1.7 | 0.2 | **17.3** |
| Jamesbond | 29.0 | 302.8 | 768.5 | 812.0 | 4703.0 | 35108.0 | **87291.7** |
| Kangaroo | 52.0 | 3035.0 | 7259.0 | 1792.0 | 15356.0 | **15487.0** | 15400.0 |
| Krull | 1598.0 | 2665.5 | 8422.3 | 10374.0 | **11447.0** | 10707.0 | 10706.8 |
| KungFuMaster | 258.5 | 22736.3 | 26059.0 | 48375.0 | 76642.0 | 73512.0 | **111138.5** |
| MontezumaRevenge | 0.0 | 4753.3 | 0.0 | 0.0 | 0.0 | 0.0 | 0.0 |
| MsPacman | 307.3 | 6951.6 | 3085.6 | 3327.3 | 5821.0 | 6349.0 | **7631.9** |
| NameThisGame | 2292.3 | 8049.0 | 8207.8 | 15572.5 | 21890.0 | **22682.0** | 16989.4 |
| Phoenix | 761.4 | 7242.6 | 8485.2 | 70324.3 | 16585.0 | 56599.0 | **174077.5** |
| Pitfall | -229.4 | 6463.7 | -286.1 | 0.0 | 0.0 | 0.0 | 0.0 |
| Pong | -20.7 | 14.6 | 19.5 | 20.9 | **21.0** | **21.0** | **21.0** |
| PrivateEye | 24.9 | 69571.3 | 146.7 | 206.0 | **350.0** | 200.0 | 140.1 |
| Qbert | 163.9 | 13455.0 | 13117.3 | 18760.3 | **572510.0** | 25750.0 | 27524.4 |
| Riverraid | 1338.5 | 17118.0 | 7377.6 | 20607.6 | 17571.0 | 17765.0 | **23560.7** |
| RoadRunner | 11.5 | 7845.0 | 39544.0 | 62151.0 | **64262.0** | 57900.0 | 58072.7 |
| Robotank | 2.2 | 11.9 | 63.9 | 27.5 | 59.4 | 62.5 | **75.7** |
| Seaquest | 68.4 | 42054.7 | 5860.6 | 931.6 | 8268.0 | **30140.0** | 29383.3 |
| Skiing | -17098.1 | -4336.9 | -13062.3 | -19949.9 | -9324.0 | -9289.0 | **-9085.3** |
| Solaris | 1236.3 | 12326.7 | 3482.8 | 133.4 | 6740.0 | **8007.0** | 6906.7 |
| SpaceInvaders | 148.0 | 1668.7 | 1692.3 | 15311.5 | 20972.0 | 28888.0 | **46498.3** |
| StarGunner | 664.0 | 10250.0 | 54282.0 | 125117.0 | 77495.0 | 74677.0 | **131981.2** |
| Tennis | -23.8 | -9.3 | 12.2 | 0.0 | **23.6** | **23.6** | 22.6 |
| TimePilot | 3568.0 | 5229.2 | 4870.0 | 7553.0 | 10345.0 | 12236.0 | **14995.2** |
| Tutankham | 11.4 | 167.6 | 68.1 | 245.9 | 297.0 | 293.0 | **309.2** |
| UpNDown | 533.4 | 11693.2 | 9989.9 | 33879.1 | 71260.0 | **88148.0** | 75474.4 |
| Venture | 0.0 | 1187.5 | 163.0 | 48.0 | 43.9 | **1318.0** | 1112 |
| VideoPinball | 16256.9 | 17667.9 | 196760.4 | 479197.0 | 705662.0 | 698045.0 | **799155.6** |
| WizardOfWor | 563.5 | 4756.5 | 2704.0 | 12352.0 | 25061.0 | 31190.0 | **44782.6** |
| YarsRevenge | 3092.9 | 54576.9 | 18098.9 | **69618.1** | 26447.0 | 28379.0 | 27691.2 |
| Zaxxon | 32.5 | 9173.3 | 5363.0 | 13886.0 | 13113.0 | **21772.0** | 15179.5 |

Table 4: Raw scores for a single seed across all games, starting with 30 no-op actions.

To align with previous works, the scores are evaluated under 30 no-op setting. As the sticky action evaluation setting proposed by Machado et al. [2018] is generally considered more meaningful in the RL community, we will add results under sticky-action evaluation setting after the conference.