[Reviews · NeurIPS 2019]

Reviewer 1



POST-REBUTTAL I thank the authors for their detailed response. My main concern was the level of experimental detail provided in the submission, and I'm pleased that the authors have committed to including more of the details implicitly contained within the code in the paper itself. My overall recommendation remains the same; I think the paper should be published, and the strong Atari results will be of interest fairly widely. However, there were a few parts of the response I wasn't convinced by: (1) "(D) Inefficient Hyperparameter": I don't agree with the authors' claim that e.g. QR-DQN requires more hyperparameters than FQF (it seems to me that both algorithmically require the number of quantiles, and the standard hyperparameters associated with network architecture and training beyond that). (2) The IQN vs. FQF discussion. In particular: (2.i) "It is not difficult to extend FQF to support arbitrary number of optimal quantiles as in IQN: we can modify the quantile network into a recurrent network so that it generates a single τ each time and takes state, action and previously outputted τ s as input to generate next τ". This seems non-trivial as a modification to the architecture, and it's not clear to me that this proposed solution would work in practice. (2.ii) "Thus, intuitively the quantiles generated in FQF should be better than sampled quantiles (as in IQN) in terms of quantile function approximation in most cases." This seems speculative, and I don't think there's anything in the paper (beyond the Atari results) to back this up. -------------------- HIGH-LEVEL COMMENTS The paper contributes a new distributional RL algorithm, which parametrizes return distributions as mixtures of Dirac deltas, and learns both the locations of the Dirac deltas, and their probability masses, extending the parametrisations used in QR-DQN and C51. This adjustment is explained reasonably clearly, and the empirical evaluations on Atari are strong, and improve considerably on related earlier algorithms. I have checked the proofs in the appendix. Based on these results, I believe the paper should be accepted. However, the paper is lacking some experimental detail, such as precise architectures/optimizers used, hyperparameter sweeps undertaken, etc.. Whilst some of this can be discerned from the attached code, not all of it can, and it any case it would be clearer if it were presented an appendix of the paper. The code itself could do with better documentation; at the moment, several “TODO” notes are left, and blocks of unused code commented out etc. In Section 4, the following claim is made “We also set a new record on the number of 222 games where RL agent performs better than human.”, with Table 1 showing that FQF beats human performance in 44 Atari games. However, the agent in (Kapturowski et al., 2019) attains super-human performance on 52 of the 57 Atari games - can the authors comment on this? DETAILED COMMENTS Lines 44-54: I think it’s inaccurate (or at least unclear) to suggest that the distribution can be better approximated by FQF than IQN. If I understand correctly, the distributions FQF can express are restricted to the parametric form in Eqn (1), whereas IQN (at test time) can sample many more quantiles than were used in each stochastic gradient computation in training, and therefore is not limited to a particular parametric family of return distributions (although clearly it is limited by the expressivity of the network). Lines 104-108: I don't agree with this statement - there will be variance in the IQN gradient introduced by using a finite number of samples in each gradient estimate, but this does not affect the approximation error of the converged system. This is similar to the variance that may occur in general distributional RL updates due to using sampled transitions rather than integrating over the dynamics of the MDP. Lines 136-138: The reason Eqn (2) cannot be minimized with SGD is that it is not possible to obtain an unbiased gradient. Line 175: It’s not clear to me what these two sentences mean. In general, the Bellman target and the fitted distribution will not be equal, due to the approximation error incurred due to the network using a parametric family of distributions. Algorithm 1: there are several minor things here that could do with neatening up: Q(s’,a’) appears outside the loop over a’; Z’ and Z are not defined. Figure 2: the error bars/mean looks off for DoubleDunk? Line 204: It is unfortunate that results for Defender and Surround were not included; however, the authors have based their experiments on a commonly-used framework, so I do not count this against the submission. The experimental results the authors report are very strong. Can the authors clarify whether the shaded regions in Figure 2 are standard deviations across the seeds run? Lines 233-238: It is unclear what the hyperparameters that the authors refer to are. If I understand correctly, their method requires selection of the hyperparameter prescribing the number of atoms to be represented (as with several other distributional RL agents), and additionally requires tuning the training hyperparameters of an additional network, since the probabilities are output by a separate network. I also disagree with the claim that it has been *theoretically* shown that the algorithm proposed in this paper achieves better performance than IQN. As I understand it, the distributions expressible by IQN and FQF are completely different: IQN can express any distribution for which the quantile function is expressible by the network. In particular, all such distributions will be absolutely continuous. In contrast, any distribution that FQF can express must necessarily be finitely-supported. Appendix: main derivations: “W_1/\partial\tau_i”->”\partial W_1/\parital\tau_i”. 2nd line: in the second integral, the lower limit should be \hat{\tau}_{i-1}. Minor typos “Dopamine” should be capitalized throughout. Line 78: “A’ \sim \pi(\cdot|x’)” ->“A’ \sim \pi(\cdot|X’)” Line 253: “Motezuma” -> “Montezuma” References Kapturowski et al.. Recurrent Experience Replay in Distributed Reinforcement Learning. ICLR 2019.

Reviewer 2



Overview: - Originality: low (incremental improvement over IQN) - Quality: medium (great results on Atari but lack of a thorough empirical analysis) - Clarity: good (overall easy to read, in spite of several notation errors) - Significance: medium (bigger numbers on Atari, but with little information as of why) Overall: I am on the fence, currently leaning towards acceptance even though I believe it would be important to better understand why the proposed algorithm works (maybe on a simpler task if Atari is too expensive). Since the full code is available, at least it should be easy enough for others to follow-up on this work to perform such investigation. Main detailed comments: - A main motivation of the paper is that "there is huge variance in the mean of the sampled quatile values in IQN", and that this hurts performance. This is somewhat surprising since the IQN paper states that "we did not find IQN to be sensitive to K, the number of samples used for the policy", which suggests that increasing K (and thus decreasing the above-mentioned variance) shouldn’t make a significant difference. This makes me wonder to which extent the improvements observed here are a result of the proposed optimization technique, vs other differences with respect to « vanilla » IQN, namely: the weighting scheme in eq. 1 (vs uniform weights in IQN), and the fact that 32 quantiles are used both for decision making (same as K=32 in IQN) and in the double-loop defining the optimized objective (vs N=N'=8 in IQN). In the absence of an ablation study, or re-running IQN with these changes, it is hard to be sure what exactly makes FQF work. - Assuming the improvements do come from using « better » quantile fractions, it would have been very interesting to investigate more carefully why this is the case exactly. In particular, currently the predicted quantile fractions are output by a network taking both the state and action as input. But is this really necessary? What if there was only the state, only the action, or even no input at all (fixed quantile fractions)? What if we simply used fixed uniformly spread quantiles for decision making, as in QR-DQN? I believe it is important to understand whether the dependence on the state and action actually matters, or if a fixed "good" set of quantile fractions is enough. Note that the authors state on l.107 that for IQN "there is no guarantee that sampled probabilities would provide better estimation than fixed uniform probabilities", but omit to verify this for their own algorithm. - The RL community has been advocating the use of sticky actions in Atari for more meaningful evaluation of RL agents. Although a comparison with old approaches on non sticky actions is definitely important for legacy reasons, an FQF vs IQN comparison with sticky actions would have strengthened the empirical validation. - I fail to understand the motivation behind minimizing the loss L_w1 (below l. 159) instead of following the gradient dW1/dtau_i computed in the Appendix proof. I can see how bringing this loss down to zero would bring the gradient to zero, but the usual practice in optimization is to follow the gradient of the loss, not minimize its square... I assume there is a good reason for this approach, but since it seems counterintuitive, it should be explained.   Minor points: - The authors use the term "probability" throughout the paper to denote a quantile fraction, as opposed to its value. This is initially confusing as it is not clear what "probability" means in this context (ex: in the abstract). To be honest I am not sure what is the best word for it: "quantile fraction" is the best I could find, but usually in distributional RL papers people just use "quantile". Anyway, I don't think "probability" should be used here since it can be misleading. - In the equation below l. 68 there should be no "Q(x, a) =" at the beginning of the equation, unless you write it for the optimal policy (and thus it should be Q*) - "Rainbow agent, current state-of-the-art in Atari-57" => specify that it's for non-distributed methods (see e.g. the chart in Fig. 2 of "Recurrent Experience Replay in Distributed Reinforcement Learning") - "In practice the number of quantiles can not be infinite and is usually set to 32": it's not clear what this number represents unless the reader is very familiar with the IQN paper (it refers to the parameter K used to choose actions). This should be clarified (ex: "In practice one needs to use a finite number of quantiles to estimate action values for decision making, e.g. 32 randomly sampled quantile fractions as in Dabney et al. [2018a]"). - The Pi^{theta, tau} below l. 126 is not defined, and it looks like tau defines a set in this equation, while in the rest of the paper tau is a single scalar => confusing notations - More generally I find this equation (below l. 126) pretty complex to parse, for a result that's actually very simple. It'd be nice to either find a simpler formulation, or/and explain it in plain English, or/and refer to a figure to show what it means. - In Fig. 1 using F_Z^-1 instead of Z to label the y axis would be more consistent with the notations of the paper - In Fig. 1 tau_N+1 on the x axis should be tau_N, and "N=5" in the caption should be "N=6" - tau_hat in Fig. 1 is not defined yet when Fig. 1 is referenced in the text (l. 135), the definition only comes on l. 147 - In equation below l. 162, w1 should be w2 (same in eq. 6 and 7) - In eq. 7, tau_hat_i should be tau_hat_j (same in the corresponding equation in Alg. 1) - w1 and w2 should be swapped in L_w1 and L_w2 in Alg. 1 - l. 190 F_w2^-1 should be F_Z,w2^-1 for notation consistency - Please clarify where the IQN training curves in Fig. 2 come from, since they do not match those from the original IQN paper, nor the numbers from Table 1 in the Appendix - Table 1 should have all « 54 » numbers in bold in rightmost column - « Our algorithm (...) is the first distributional RL algorithm removing all inefficient hyper-parameters that require manual tuning previously » => the number of quantiles still needs to be tuned, as well as the quantile embedding parameters (equation below l. 187), and the quantile proposal network architecture... - « we believe that our algorithm presents a whole new direction for distributional RL » => I believe this sentence should be toned down, as this paper is an incremental improvement over IQN, not a « whole new direction ». Note also that regarding the question « is it possible that we leverage the distribution and develop an exploration-aware algorithm? », the short answer is « yes » since such work already exists (Google up « distributional reinforcement learning exploration » for instance) - In Appendix eq. 1 the sum over i should be from 0 to N-1 - In Appendix proof, beginning of equations, W1/dtau_i should be dW1/dtau_i - In Appendix proof, using large parentheses $\left(...\right)$ would make some equations easier to parse - In Appendix, second line of the equations in the proof, the second integral should start from tau_hat_i-1, not tau_hat_i - In Appendix, the derivation of eq. 3 is not detailed enough in my opinion, I had to write it down to understand it. Since this is the Appendix, there should be enough room to spell it out more explicitly. There is also either a mistake or (more likely) an abuse of notation when writing e.g. d/dtau_i F_Z^-1(tau_hat_i-1) since it should be dF_Z^-1/dtau taken at tau=tau_hat_i-1 - In Appendix, last too equalities in the proof equations, tau_hat_i+1 should be tau_hat_i   English typos: - "network(IQN)" missing space (x2) - "X and A denotes" denote - "while agent interacts" the agent - "C51 out-perform" outperforms - "is able to approximate to the full quantile" remove 2nd 'to' - "with infinity number of quantiles" with infinitely many quantiles - l. 150 remove "for" - « element-wise(Hadamard) » missing space - « Motezuma » Update after author feedback: Thank you for the response, I appreciate the many clarifications that were made, especially regarding the motivation for minimizing L_w1 -- I definitely encourage you to "figure out why and explain it in detail" :) [feedback l. 36] However, the other important concerns I had regarding the empirical analysis of the proposed algorithm were only partially addressed (through the promise of extra experiments -- whose results are still uncertain) [feedback l. 9, 25-28] I still believe that this paper deserves being published, mostly due to the impressive results on Atari, so I will keep my original recommendation (6). I hope that the final version will contain relevant additional experiments to better explain why the proposed technique works so well!

Reviewer 3



Originality: This work introduces a novel method for optimizing the selection of quantiles to minimize error in the return distribution. This method is then combined with IQN to produce a new RL algorithm that improves the ability of a network to accurately estimate the return distribution. Quality: The paper makes the following claims: 1) “Self-adjusting probabilities can approximate the true distribution better than fixed or sampled probabilities” -- line 53 2) “we believe that the former one would achieve better results since it approximates the quantile function with a stair case function, instead of viewing the process as sampling which leads to instability”-- line 194 3) “It shows that FQF out-performed all existing distributional RL algorithms” -- line 219 4)“FQF is generally much faster and stabler thanks to self-adjusting probabilities” -- line 224 In trying to evaluate these claims I have questions about their scope and justification. 1) In Figure 1, it is shown for some distribution optimizing the choice of quantiles can provide a better approximation to the distribution. However, the contribution of this paper is a method to better estimate the return distribution, but no experiment is conducted that shows how well the introduced method approximates the return distribution compared to previous methods. 2) Are there unreported experiments that show “sampling leads to instability”. This would be a nice small contribution to add to the paper that could influence future research. 3) The experiments were conducted with 3 trials. With the known high variance of performance over random trials how certain is the result that FQF is better than other distributional RL algorithms? Reproducibility worksheet indicates that there is a clear description of the central tendency and variation, but I do not see mention of the variance or uncertainty of the performance measurements. The error bands on the learn curve plots are also undefined. Additionally, FQF used the same hyperparameters as IQN. This comparison only shows that FQF obtained higher performance for these hyperparameters. Does this imply that FQF will always outperform IQN with the same hyperparameters? I believe the result needs to be qualified with the uncertainty of the performance estimate and to what extent the experimental data shows FQN being greater than the other algorithms. 4) This claim has two parts: faster and more stable. I assume (please correct me if I am wrong) that faster is with respect to how quickly the algorithm learns. Under what measure is the speed of each algorithm being compared? Similarly, for stability what is the measure that is being used to compare these algorithms? In my reading of the paper, I developed other questions that I could not find answers for and that would improve the quality of the paper if answered. If actions are selected via expectations, then does it matter how well the distribution is approximated so long as the mean is preserved? How does minimizing the error in the approximated distribution affect the approximation of the mean? It is mentioned that “learning rates are relatively small compared with the quantile network to keep the probabilities relatively stable while training”. How stable is this algorithm to the relative setting of learning rates? Is selecting hyperparameters easy? "First, does the 1-Wasserstein error converge to its minimal value when we minimize the surrogate function derived by its derivative? While the answer is yes on some toy problems we test with fixed quantile functions," This toy problem would be great to see! It would add a lot to the paper to see how well the distribution is approximated. This is the critical claim of the paper. Experiments demonstrating this are more valuable than getting SOTA performance results on Atari. These experiments could also be conducted on domains that require less computational than Atari. The discussion about the increased computational cost of the new method is appreciated (lines 228-231). Clarity: some minor notes to help the clarity of the writing Add mention that there is a proof of Proposition 1 in the appendix. In the discussion section, it states that: “Our algorithm, by the time of writing, is the first distributional RL algorithm removing all inefficient hyper-parameters that require manual tuning previously. What does “inefficient hyperparameters” mean? I am looking for a precise clarification of this statement as to which hyperparameters are removed. Significance: The main idea presented in this paper of optimizing the selection of quantiles could have significant impacts for distributional RL. Answering the question “How important is it to have an accurate approximation of return to the performance of distributional RL algorithms?” would provide a valuable contribution to the investigation and development of new distributional RL algorithms. This paper attempts to answer how performance is impacted but does not address to what extent the accuracy of the approximated distribution impacted performance. Thus, the knowledge generated by this paper is limited to the presentation of the idea and a performance measure. These are less actionable than furthering the understanding of how to best approximate the return distribution. ---Update I give my thanks to the authors for their clear response. I think my question about to what degree does having an accurate estimate of the distribution affect performance was not conveyed correctly. It was not about comparing distributional RL methods to regular, but about how accurate does the distribution approximation needs to be? If one can use say 3 quantiles versus 30 and still select the optimal action, then having a close approximation does not matter. Unless it allows for better representation learning and quicker policy improvement. Nevertheless, I appreciate the experiments detailing that FQF did learn a better approximation to the return distribution than some methods and updated my score. Please include these experiments with IQN and a description in the next version of the paper.

[Author Response · NeurIPS 2019]

*General response:* Thanks a lot for your comments and suggestions! We will fix typos, add more details and re-organize
the paper to improve clarity.

Figure 1                     Figure 2                     Figure 3

**(A)** *Toy Experiment for Distribution Approximation:* Figure 1 and 2 are toy cases of quantile function approximation
using C51, QR-DQN and FQF (IQN should be viewed as sampling in a distribution instead of approximating it). $W_1$
loss suggests that FQF does have advantage in approximating distribution and thus achieve better estimation on values.

**(B)** *Experiment Details:* Figure 3 shows the general architecture of FQF. The learning rate of the probability network is
set to 0.0001 but as long as its smaller than 0.001 FQF works quite well. We commit to add more experimental details in
the camera-ready version. **(C)** *More Experiment under Different Settings:* We commit to compare FQF with IQN under
different set of hyperparameters for ablative analysis in the camera-ready version. **(D)** *Inefficient Hyparameter:* By
'inefficient hyperparameter' we mean the atom locations (uniformly distributed between -10 and 10) in C51, quantiles
locations in QR-DQN and sampled quantiles locations in IQN (it still requires a distribution to sample from). Note that
FQF requires only the number of quantiles. **(E)** *State-of-Art Performance:* Thanks for pointing out [Kapturowski et al.,
2019]! Yes, FQF is the best among single-actor algorithms, i.e., non-distributed methods. We will clarify this in the
new version and make our claim more accurate. Besides, we believe that combining advantages of distributional RL
and distributed RL is an exciting direction for further research.

*To reviewer 1:* **(I)** *Experiment Details:* Please refer to **(B)** in general response. **(II)** *IQN v.s. FQF:* IQN does have the
advantage that it can sample as many quantiles as possible, but with the same number of quantiles FQF gives much
more accurate approximations. It is not difficult to extend FQF to support arbitrary number of optimal quantiles as
in IQN: we can modify the quantile network into a recurrent network so that it generates a single $\tau$ each time and
takes state, action and previously outputted $\tau$s as input to generate next $\tau$. FQF finds quantiles that can most efficiently
express a distribution and computes the expectation of Q with such expression. Thus, intuitively the quantiles generated
in FQF should be better than sampled quantiles (as in IQN) in terms of quantile function approximation in most cases.
We leave the theoretical analysis and comparison between IQN and FQF to future work.

*To reviewer 2:* **(I)** *More Experiment under Different Settings:* Please refer to **(C)** in general response. We will add
experiments regarding different K and N in the new version. It is a great idea to investigate what impacts the quantiles
generated in FQF, and we will include the related results in the appendix of the new version as well as the results
using sticky actions. **(II)** *Different Weights v.s. Uniform Weights:* We actually emailed the authors of IQN regarding
the weighting scheme and tested it. We all found that using different weights or uniform weights does not really
impact performance. **(III)** *The Motivation for Minimizing $L_w1$:* Why we minimize the square of the gradient instead of
following the gradient, we agree that it should be explained. Although we cannot compute $W_1$, we derived its derivative
for $\tau$ in the appendix and directly following the gradient is the most intuitive way to minimize $W_1$. Unfortunately, we
found following the gradient makes the probability proposal network very hard to train at the beginning when quantile
outputs are unstable. It often causes $\tau$s to converge to something like (0,0,0,1,1,1,1,1), while using square (not very
different from following the gradient actually) performs much better. This may be a result of the cumulative softmax
architecture. We will check if we can figure out why and explain it in detail.

*To reviewer 3:* **(I)** *How well the distribution is approximated/Inefficient Hyper-parameters::* Please refer to **(A)/(D)** in
the general response. **(II)** *Supporting Experiments:* We agree with you on adding supporting experiments other than
performance, some of those experiments are shown in the general response section and we will add more to the new
version. We also agree that performance measurements on variance is important for further distributional RL research
and should be included. **(III)** *Why distribution matters?* The reason why distribution matters had been studied by
several previous works: 1). An Analysis of Categorical Distributional Reinforcement Learning (Rowland et al. 2018)
2). As Expected? An Analysis of Distributional Reinforcement Learning (Lyle and Bellemare, 2018). The general
conclusion is that "the more complex the setting the less likely it is that distributional and expected RL algorithms
will behave in the same way". So it is necessary for distributional RL algorithms to approximate the full distribution,
even when we only need its mean. As shown in our toy case, FQF does achieve better distribution approximation. **(IV)**
*"faster and more stable"* We apologize for the misleading choice of word in "faster and more stable". What we actually
meant by 'faster' was higher sampling efficiency compared with IQN instead of training speed, and by stability we refer
to smaller error bands. We will add numerical results to support this claim in the new version.

[Meta-Review · NeurIPS 2019]

The reviewers expressed some concerns about the significance of the paper, given that the main contribution is a SOTA result. However, they conclude that the Atari benchmark is sufficiently mature that an increase in this direction is of general interest. Some of the sticking points that should be addressed in the revision are: 1) consider performing additional empirical analysis to better understand how the method operates, 2) include further details (as requested by the reviewers).